# Few-Shot Parameter-Efficient Fine-Tuning is Better and Cheaper than In-Context Learning

**Haokun Liu**[*]   **Derek Tam**[*]   **Mohammed Muqeeth**[*]

**Jay Mohta**   **Tenghao Huang**   **Mohit Bansal**   **Colin Raffel**

Department of Computer Science
University of North Carolina at Chapel Hill
{haokunl,dtredsox,muqeeth,craffel}@cs.unc.edu

## Abstract

Few-shot in-context learning (ICL) enables pre-trained language models to perform a previously-unseen task without any gradient-based training by feeding a small number of training examples as part of the input. ICL incurs substantial computational, memory, and storage costs because it involves processing all of the training examples every time a prediction is made. Parameter-efficient fine-tuning (PEFT) (e.g. adapter modules, prompt tuning, sparse update methods, etc.) offers an alternative paradigm where a small set of parameters are trained to enable a model to perform the new task. In this paper, we rigorously compare few-shot ICL and PEFT and demonstrate that the latter offers better accuracy as well as dramatically lower computational costs. Along the way, we introduce a new PEFT method called $(IA)^3$ that scales activations by learned vectors, attaining stronger performance while only introducing a relatively tiny amount of new parameters. We also propose a simple recipe based on the T0 model [1] called `T-Few` that can be applied to new tasks without task-specific tuning or modifications. We validate the effectiveness of `T-Few` on completely unseen tasks by applying it to the RAFT benchmark [2], attaining super-human performance for the first time and outperforming the state-of-the-art by 6% absolute. All of the code used in our experiments is publicly available.[1]

## 1   Introduction

Pre-trained language models have become a cornerstone of natural language processing, thanks to the fact that they can dramatically improve *data efficiency* on tasks of interest – i.e., using a pre-trained language model for initialization often produces better results with less labeled data. A historically common approach has been gradient-based fine-tuning on a downstream task of interest with pre-trained parameters as the initialization. While fine-tuning has produced many state-of-the-art results [1], it results in a model specialized for a single task with an entirely new set of parameter values, which can become impractical when fine-tuning on many downstream tasks.

An alternative approach popularized by [3, 4] is *in-context learning* (ICL), which induces a model to perform a downstream task by inputting *prompted* examples. Few-shot prompting converts a small collection of input-target pairs into (typically) human-understandable instructions and examples [3, 4], along with a single unlabeled example for which a prediction is desired. Notably, ICL requires no gradient-based training and therefore allows a single model to immediately perform a wide variety of tasks. Performing ICL therefore solely relies on the capabilities that a model learned during pre-training. These characteristics have led to a great deal of recent interest in ICL methods [5–10].

---

[*]Equal contribution.

[1] https://github.com/r-three/t-few

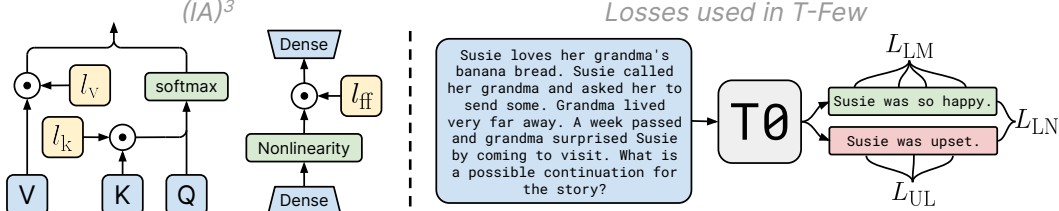

Figure 1: Diagram of $(IA)^3$ and the loss terms used in the T-Few recipe. *Left:* $(IA)^3$ introduces the learned vectors $l_k, l_v$, and $l_{ff}$ which respectively rescale (via element-wise multiplication, visualized as $\odot$) the keys and values in attention mechanisms and the inner activations in position-wise feed-forward networks. *Right:* In addition to a standard cross-entropy loss $L_{LM}$, we introduce an unlikelihood loss $L_{UL}$ that lowers the probability of incorrect outputs and a length-normalized loss $L_{LN}$ that applies a standard softmax cross-entropy loss to length-normalized log-probabilities of all output choices.

Despite the practical benefits of ICL, it has several major drawbacks. First, processing all prompted input-target pairs every time the model makes a prediction incurs significant compute costs. Second, ICL typically produces inferior performance compared to fine-tuning [4]. Finally, the exact formatting of the prompt (including the wording [11] and ordering of examples [12]) can have significant and unpredictable impact on the model's performance, far beyond inter-run variation of fine-tuning. Recent work has also demonstrated that ICL can perform well even when provided with incorrect labels, raising questions as to how much learning is taking place at all [9].

An additional paradigm for enabling a model to perform a new task with minimal updates is *parameter-efficient fine-tuning* (PEFT), where a pre-trained model is fine-tuned by only updating a small number of added or selected parameters. Recent methods have matched the performance of fine-tuning the full model while only updating or adding a small fraction (e.g. 0.01%) of the full model's parameters [13, 14]. Furthermore, certain PEFT methods allow *mixed-task batches* where different examples in a batch are processed differently [14], making both PEFT and ICL viable for multitask models.

While the benefits of PEFT address some shortcomings of fine-tuning (when compared to ICL), there has been relatively little focus on whether PEFT methods work well when very little labeled data is available. Our primary goal in this paper is to close this gap by proposing a recipe – i.e., a model, a PEFT method, and a fixed set of hyperparameters – that attains strong performance on novel, unseen tasks while only updating a tiny fraction of the model's parameters. Specifically, we base our approach on the T0 model [1], a variant of T5 [15] fine-tuned on a multitask mixture of prompted datasets. To improve performance on classification and multiple-choice tasks, we add unlikelihood [16, 17] and length normalization-based [4] loss terms. In addition, we develop $(IA)^3$, a PEFT method that multiplies intermediate activations by learned vectors. $(IA)^3$ attains stronger performance than full-model fine-tuning while updating up to $10,000\times$ fewer parameters. Finally, we demonstrate the benefits of pre-training the $(IA)^3$ parameters before fine-tuning [18, 19]. Our overall recipe, which we dub "T-Few", performs significantly better than ICL (even against $16\times$ larger models) and outperforms humans for the first time on the real-world few-shot learning benchmark RAFT [2] while requiring dramatically less compute and allowing for mixed-task batches during inference. To facilitate the use of T-Few on new problems and future research on PEFT, we release our code.[1]

After providing background on ICL and PEFT in the following section, we discuss the design of T-Few in section 3. In section 4, we present experiments comparing T-Few to strong ICL baselines. Finally, we discuss related work in section 5 and conclude in section 6.

## 2 Background

In this section, we provide am verview of ICL and PEFT with a focus on characterizing the computation, memory, and on-disk storage costs of making a prediction. Real-world costs depend on implementation and hardware, so we report costs in terms of FLOPs for computation and bytes for memory and storage, respectively. Additional related work is discussed in section 5.

## 2.1 Few-shot in-context learning (ICL)

ICL [3, 4] aims to induce a model to perform a task by feeding in concatenated and prompted input-target examples (called "shots") along with an unlabeled query example. Taking the cycled letter task from Brown et al. [4] as an example, a 4-shot input or *context* would be "`Please unscramble the letters into a word, and write that word:  asinoc = casino, yfrogg = froggy, plesim = simple, iggestb = biggest, astedro =`", for which the desired output would be "`roasted`". ICL induces an autoregressive language model to perform this task by feeding in the context and sampling from the model. For classification tasks, each label is associated with a string (e.g. "`positive`" and "`negative`" for sentiment analysis) and a label is assigned by choosing the label string that the model assigns the highest probability to. For multiple-choice tasks (e.g. choosing between $N$ possible answers to a question), the model's prediction is similarly determined by determining which choice is assigned the highest probability.

The primary advantage of ICL is that it enables a single model to perform many tasks immediately without fine-tuning. This also enables *mixed-task batches*, where different examples in a batch of data correspond to different tasks by using different contexts in the input. ICL is also typically performed with only a limited number of labeled examples – called few-shot learning – making it data-efficient.

Despite these advantages, ICL comes with significant practical drawbacks: First, making a prediction is dramatically more expensive because the model needs to process all of the in-context labeled examples. Specifically, ignoring the quadratic complexity of self-attention operations in Transformer language models (which are typically small compared to the costs of the rest of the model [20]), processing the $k$ training examples for $k$-shot ICL increases the computational cost by approximately $k + 1$ times compared to processing the unlabeled example alone. Memory costs similarly scale approximately linearly with $k$, though during inference the memory costs are typically dominated by storing the model's parameters. Separately, there is a small amount of on-disk storage required for storing the in-context examples for a given task. For example, storing 32 examples for a task where the prompted input and target for each example is 512 tokens long would require about 66 kilobytes of storage on disk (32 examples $\times$ 512 tokens $\times$ 32 bits).

Beyond the aforementioned costs, ICL also exhibits unintuitive behavior. Zhao et al. [12] showed that the *ordering* of examples in the context heavily influences the model's predictions. Min et al. [9] showed that ICL can still perform well even if the in-context example labels are swapped (i.e. made incorrect), which raises questions about whether ICL is really "learning" from the labeled examples.

Various approaches have been proposed to mitigate these issues. One way to decrease computational costs is to cache the key and value vectors for in-context examples. This is possible because decoder-only Transformer language models have a causal masking pattern, so the model's activations for the context do not do not depend on the unlabeled example. In an extreme case, 32-shot ICL with 512 tokens per in-context example would result in over 144 gigabytes of cached key and value vectors for the GPT-3 model (32 examples $\times$ 512 tokens $\times$ 96 layers $\times$ 12288 $d_{model}$ $\times$ 32 bits *each* for the key and value vectors). Separately, Min et al. [21] proposed *ensemble ICL*, where instead of using the output probability from concatenating the $k$ training examples, the output probabilities of the model on each training example (i.e. 1-shot ICL for each of the $k$ examples) are multiplied together. This lowers the non-parameter memory cost by a factor of $k/2$ but increases the computational cost by a factor of 2. In terms of task performance, Min et al. [21] find that ensemble ICL outperforms the standard concatenative variant.

## 2.2 Parameter-efficient fine-tuning

While standard fine-tuning updates all parameters of the pre-trained model, it has been demonstrated that it is possible to instead update or add a relatively small number of parameters. Early methods proposed adding *adapters* [22–24], which are small trainable feed-forward networks inserted between the layers in the fixed pre-trained model. Since then, various sophisticated PEFT methods have been proposed, including methods that choose a sparse subset of parameters to train [25, 26], produce low-rank updates [13], perform optimization in a lower-dimensional subspace [27], add low-rank adapters using hypercomplex multiplication [28], and more. Relatedly, *prompt tuning* [14] and *prefix tuning* [29] concatenate learned continuous embeddings to the model's input or activations to induce it to perform a task; this can be seen as a PEFT method [30]. State-of-the-art PEFT methods can

match the performance of fine-tuning all of the model's parameters while updating only a tiny fraction (e.g. 0.01%) of the model's parameters.

PEFT drastically reduces the memory and storage requirements for training and saving the model. In addition, certain PEFT methods straightforwardly allow mixed-task batches – for example, prompt tuning enables a single model to perform many tasks simply by concatenating different prompt embeddings to each example in the batch [14]. On the other hand, PEFT methods that re-parameterize the model (e.g. [27, 13]) are costly or onerous for mixed-task batches. Separately, different PEFT methods increase the computation and memory required to perform inference by different amounts. For example, adapters effectively add additional (small) layers to the model, resulting in small but non-negligible increases in computational costs and memory. An additional cost incurred by PEFT is the cost of fine-tuning itself, which must be performed once and is then amortized as the model is used for inference. However, we will show that PEFT can be dramatically more computationally efficient when considering both fine-tuning and inference while achieving better accuracy than ICL.

## 3    Designing the `T-Few` Recipe

Given that PEFT allows a model to be adapted to a new task with relatively small storage requirements and computational cost, we argue that PEFT presents a promising alternative to ICL. Our goal is therefore to develop a recipe that allows a model to attain high accuracy on new tasks with limited labeled examples while allowing mixed-task batches during inference and incurring minimal computational and storage costs. By *recipe*, we mean a specific model and hyperparameter setting that provides strong performance on any new task without manual tuning or per-task adjustments. In this way, we can ensure that our approach is a realistic option in few-shot settings where limited labeled data is available for evaluation [31, 32].

### 3.1    Model and Datasets

As a first step, we must choose a pre-trained model. Ideally, the model should attain high performance on new tasks after fine-tuning on a limited number of labeled examples. In preliminary experiments applying PEFT methods to different pre-trained models, we attained the best performance with T0 [1]. T0 is based on T5 [15], an encoder-decoder Transformer model [33] that was pre-trained via a masked language modeling objective [34] on a large corpus of unlabeled text data. T0 was created by fine-tuning T5 on a multitask mixture of datasets in order to enable zero-shot generalization, i.e. the ability to perform tasks without any additional gradient-based training. Examples in the datasets used to train T0 were prompted by applying the prompt templates from the Public Pool of Prompts (P3 [35]), which convert each example in each dataset to a prompted text-to-text format where each label corresponds to a different string. For brevity, we omit a detailed description of T0 and T5; interested readers can refer to Sanh et al. [1] and Raffel et al. [15]. T0 was released in three billion and eleven billion parameter variants, referred to as "T0-3B" and simply "T0" respectively. In this section (where our goal is to design the `T-Few` recipe through extensive experimentation), we use T0-3B to reduce computational costs. For all models and experiments, we use Hugging Face Transformers [36].

While T0 was designed for zero-shot generalization, we will demonstrate that it also attains strong performance after fine-tuning with only a few labeled examples. To test T0's generalization, Sanh et al. [1] chose a set of tasks (and corresponding datasets) to hold out from the multitask training mixture – specifically, sentence completion (COPA [37], H-SWAG [38], and Story Cloze [39] datasets), natural language inference (ANLI [40], CB [41], and RTE [42]), coreference resolution (WSC [43] and Winogrande [44]), and word sense disambiguation (WiC [45]). Evaluation of generalization capabilities can then be straightforwardly done by measuring performance on these held-out datasets. We also will later test `T-Few`'s abilities in the RAFT benchmark [2] in section 4.3, a collection of unseen "real-world" few-shot tasks with no validation set and a held-out test set. ANLI, WiC, WSC is licensed under a Creative Commons License. Winogrande is licensed under an Apache license. COPA is under a BSD-2 Clause license. We could not find the license of RTE and CB but they are part of SuperGLUE which mentions the datasets are allowed for use in research context.

To ease comparison, we use the same number of few-shot training examples for each dataset as Brown et al. [4], which varies from 20 to 70. Unfortunately, the few-shot dataset subsets used by Brown et al. [4] have not been publicly disclosed. To allow for a more robust comparison, we therefore constructed five few-shot datasets by sampling subsets with different seeds and report the median and

interquartile range. We prompt examples using a randomly-sampled prompt template from P3 Bach et al. [35] for each example at each step. Unless otherwise stated, we train our model for 1K steps with a batch size of 8 and report performance at the end of training.

For evaluation, we use "rank classification", where the model's log-probabilities for all possible label strings are ranked and the model's prediction is considered correct if the highest-ranked choice is the correct answer. Rank classification evaluation is compatible with both classification and multiple-choice tasks. Since model performance can vary significantly depending on the prompt template used, we report the median accuracy across all prompt templates from P3 and across few-shot data subsets for each dataset. For all datasets, we report the accuracy on the test set or validation set when the test labels are not public (e.g. SuperGLUE datasets). In the main text, we report median accuracy across the nine datasets mentioned above. Detailed results on each dataset are provided in the appendices.

## 3.2 Unlikelihood Training and Length Normalization

Before investigating PEFT methods, we first explore two additional loss terms to improve the performance of few-shot fine-tuning of language models. Language models are normally trained with cross-entropy loss $L_{\mathrm{LM}} = -\frac{1}{T} \sum_t \log p(y_t|\mathbf{x}, y_{<t})$ where the model is trained to increase the probability of the correct target sequence $\mathbf{y} = (y_1, y_2, \ldots, y_T)$ given the input sequence $\mathbf{x}$.

For evaluation, we use rank classification (described in section 3.1) which depends on both the probabilities that the model assigns to the correct choice as well as the incorrect choices. To account for this during training, we add an unlikelihood loss [16, 17]:

$$L_{\mathrm{UL}} = -\frac{\sum_{n=1}^{N} \sum_{t=1}^{T^{(n)}} \log(1 - p(\hat{y}_i^{(n)}|\mathbf{x}, \hat{y}_{<t}^{(n)}))}{\sum_{n=1}^{N} T^{(n)}} \tag{1}$$

which discourages the model from predicting tokens from incorrect target sequences, where $\hat{\mathbf{y}}^{(n)} = (\hat{y}_1, \hat{y}_2, \ldots, \hat{y}_{T^{(n)}})$ is the $n$-th of $N$ incorrect target sequences. We hypothesize that adding $L_{\mathrm{UL}}$ will improve results on rank classification because the model will be trained to assign lower probabilities to incorrect choices, thereby improving the chance that the correct choice is ranked highest.

The possible target sequences for a given training example can have significantly different lengths, especially in multiple-choice tasks. Ranking each choice based on probability can therefore "favor" shorter choices because the model's assigned probability to each token is $\leq 1$. To rectify this, we consider using length normalization when performing rank classification, which divides the model's score on each possible answer choice by the number of tokens in the choice (as used in GPT-3 [4]). When using length normalization during evaluation, we introduce an additional loss term during training that more closely reflects length-normalized evaluation. First, we compute the length-normalized log probability of a given output sequence $\beta(\mathbf{x}, \mathbf{y}) = \frac{1}{T} \sum_{t=1}^{T} \log p(y_t|\mathbf{x}, y_{<t})$. Then, we maximize the length-normalized log probability of the correct answer choice by minimizing the softmax cross-entropy loss:

$$L_{\mathrm{LN}} = -\log \frac{\exp(\beta(\mathbf{x}, \mathbf{y}))}{\exp(\beta(\mathbf{x}, \mathbf{y})) + \sum_{n=1}^{N} \exp(\beta(\mathbf{x}, \hat{\mathbf{y}}^{(n)}))} \tag{2}$$

When training a model with $L_{\mathrm{LM}}$, $L_{\mathrm{UL}}$, and $L_{\mathrm{LN}}$, we simply sum them. This avoids introducing any hyperparameters that would be problematic to tune in the few-shot setting (where realistically-sized validation sets are tiny by necessity [31, 32]).

We report the results of fine-tuning all of T0-3B's parameters with and without length normalization on all datasets in appendix B. We find that adding $L_{\mathrm{LN}}$ improves the accuracy from 60.7% to 62.71% and including both $L_{\mathrm{UL}}$ and $L_{\mathrm{LN}}$ provides a further improvement to 63.3%. Since these loss terms improve performance without introducing any additional hyperparameters, we include them in our recipe and use them in all following experiments.

## 3.3 Parameter-efficient fine-tuning with (IA)³

In order to compare favorably to few-shot ICL, we need a PEFT method that has the following properties: First, it must add or update as few parameters as possible to avoid incurring storage and memory costs. Second, it should achieve strong accuracy after few-shot training on new tasks.

Finally, it must allow for mixed-task batches, since that is a capability of ICL. In order to easily enable mixed-task batches, a PEFT method should ideally not modify the model itself. Otherwise, each example in a batch would effectively need to be processed by a different model or computational graph. A more convenient alternative is provided by methods that directly modify the *activations* of the model since this can be done independently and cheaply to each example in the batch according to which task the example corresponds to. Prompt tuning and prefix tuning methods [14, 29] work by concatenating learned vectors to activation or embedding sequences and are therefore examples of activation-modifying PEFT methods that allow for mixed-task batches. However, as we will discuss later, we were unable to attain reasonable accuracy with prompt tuning and found that the more performant PEFT methods did not allow for mixed-task batches. We therefore developed a new PEFT method that meets our desiderata.

As an alternative, we explored element-wise multiplication (i.e. rescaling) of the model's activations against a learned vector. Specifically, we consider adaptation of the form $l \odot x$ where $l \in \mathbb{R}^d$ is a learned task-specific vector, $\odot$ represents element-wise multiplication, and $x \in \mathbb{R}^{T \times d}$ is a length-$T$ sequence of activations. We use "broadcasting notation" [46] so that the $(i, j)^{\text{th}}$ entry of $l \odot x$ is $l_j x_{i,j}$. In preliminary experiments, we found it was not necessary to introduce a learned rescaling vector for each set of activations in the Transformer model. Instead, we found it was sufficient to introduce rescaling vectors on the keys and values in self-attention and encoder-decoder attention mechanisms and on the intermediate activation of the position-wise feed-forward networks. Specifically, using the notation from Vaswani et al. [33], we introduce three learned vectors $l_{\text{k}} \in \mathbb{R}^{d_{\text{k}}}, l_{\text{v}} \in \mathbb{R}^{d_{\text{v}}}$, and $l_{\text{ff}} \in \mathbb{R}^{d_{\text{ff}}}$, which are introduced into the attention mechanisms as:

$$\text{softmax}\left(\frac{Q(l_{\text{k}} \odot K^T)}{\sqrt{d_k}}\right)(l_{\text{v}} \odot V)$$

and in the position-wise feed-forward networks as $(l_{\text{ff}} \odot \gamma(W_1 x))W_2$, where $\gamma$ is the feed-forward network nonlinearity. We introduce a separate set of $l_{\text{k}}, l_{\text{v}}$, and $l_{\text{ff}}$ vectors in each Transformer layer block. This adds a total of $L(d_k + d_v + d_{\text{ff}})$ new parameters for a $L$-layer-block Transformer encoder and $L(2d_k + 2d_v + d_{\text{ff}})$ (with factors of 2 accounting for the presence of both self-attention and encoder-decoder attention) for a $L$-layer-block decoder. $l_{\text{k}}, l_{\text{v}}$, and $l_{\text{ff}}$ are all initialized with ones so that the overall function computed by the model does not change when they are added. We call our method (IA)³, which stands for "Infused Adapter by Inhibiting and Amplifying Inner Activations".

(IA)³ makes mixed-task batches possible because each sequence of activations in the batch can be separately and cheaply multiplied by its associated learned task vector. We also note that, in the event that a model will only be used on a single task, the modifications introduced by (IA)³ can also be applied to weight matrices permanently so that no elementwise multiplication is required and the model's architecture remains unchanged. This possible because element-wise multiplications performed in (IA)³ always co-occur with a matrix multiplication, and $l \odot Wx = (l \odot W)x$. In this case, our method incurs no additional computational cost compared to the original model.

To validate (IA)³, we compare it to a large variety of existing adaptation methods in our setting of fine-tuning T0-3B on few-shot datasets from held-out tasks. Specifically, we compare with 9 strong PEFT methods: BitFit [47] which updates only the bias parameters; Adapters [23] which introduce task-specific layers after the self-attention and position-wise feed-forward networks; Compacter and Compacter++ [28] which improve upon adapters by using low-rank matrices and hypercomplex multiplication; prompt tuning [14] which learns task-specific prompt embeddings that are concatenated to the model's input; FISH Mask [26] which chooses a subset of parameters to update based on their approximate Fisher information; Intrinsic SAID [27] which performs optimization in a low-dimensional subspace; prefix-tuning [29] which learns task-specific vectors that are concatenated to the model's activations; and LoRA [13] which assigns low-rank updates to parameter matrices. Additionally, we include the baselines of full-model fine-tuning and updating only the layer normalization parameters. For certain methods that allow changing the parameter efficiency, we report results for different budgets: 0.2% and 0.02% sparsity for FISH Mask, 10 and 100 learned prompt vectors for prompt tuning, and 20,000- or 500,000-dimensional subspaces for Intrinsic SAID.

The results are shown in fig. 2, with detailed per-dataset results in appendix C. We find that (IA)³ is the only method that attains higher accuracy than the full-model-fine-tuning baseline. While other PEFT methods (e.g. Intrinsic SAID and prompt tuning) update or introduce fewer parameters, (IA)³ performs considerably better. Our results and setting differ with some past work on the PEFT methods we compare against. Mahabadi et al. [28] report that Compacter and Compacter++

outperform full-model fine-tuning, including in the few-shot setting. Lester et al. [14] found that prompt tuning could match full-model fine-tuning, and in subsequent work Wei et al. [48] found that prompt tuning performed well when applied to a multitask fine-tuned model in the few-shot setting. In both cases, we experimented with various hyperparameter choices to try to match past results. We hypothesize the disagreement comes from us using a different model and different datasets. For prompt tuning specifically, we noticed that the validation set performance could fluctuate wildly over the course of training, hinting at possible optimization issues.

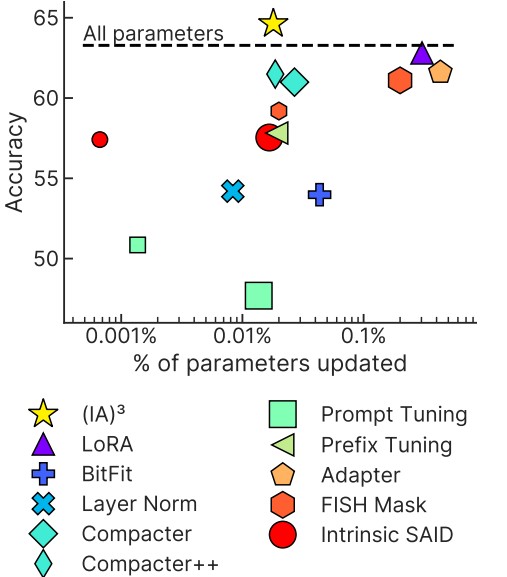

Figure 2: Accuracy of PEFT methods with $L_{\text{UL}}$ and $L_{\text{LN}}$ when applied to T0-3B. Methods that with variable parameter budgets are represented with larger and smaller markers for more or less parameters.

Figure 3: Accuracy of different few-shot learning methods. `T-Few` uses `(IA)³` for PEFT methods of T0, T0 uses zero-shot learning, and T5+LM and the GPT-3 variants use few-shot ICL. The x-axis corresponds to inference costs; details are provided in section 4.2.

## 3.4 Pre-training `(IA)³`

In recent work, Gu et al. [18], Vu et al. [19] showed that *pre-training* the prompt embeddings in prompt tuning can improve performance when fine-tuning on downstream few-shot tasks. For pre-training, Gu et al. [18] use a suite of self-supervised tasks applied to unlabeled text data, and Vu et al. [19] consider using embeddings from a separate task or multitask mixture. We follow Vu et al. [19] and simply pre-train the new parameters introduced by `(IA)³` on the same multitask mixture used to train T0. We pre-train for 100,000 steps with a batch size of 16 before fine-tuning the `(IA)³` parameters on each individual downstream dataset. A full comparison of accuracy with and without pre-training `(IA)³` is detailed in appendix D. We find that pre-training improves fine-tuned accuracy from 64.6 to 65.8 and therefore add it to our recipe.

## 3.5 Combining the ingredients

In summary, the `T-Few` recipe is defined as follows: We use the T0 model as a backbone. We add `(IA)³` for downstream task adaptation and use parameters initialized from pre-training `(IA)³` on the same multitask mixture for T0. As an objective, we use the sum of a standard language modeling loss $L_{\text{LM}}$, an unlikelihood loss $L_{\text{UL}}$ for incorrect choices, and a length-normalized loss $L_{\text{LN}}$. We train for 1,000 steps with a batch size of 8 sequences using the Adafactor optimizer [49] with a learning rate of $3e^{-3}$ and a linear decay schedule with a 60-step warmup. We apply prompt templates to downstream datasets during training and inference to convert each example into an instructive text-to-text format. Importantly, *we apply this recipe to every downstream dataset in exactly the same way* without per-dataset hyperparameter tuning or modifications. This makes the recipe a realistic option for few-shot learning settings where validation sets are tiny by definition [31, 32].

# 4 Outperforming ICL with `T-Few`

Having designed and established the `T-Few` recipe on T0-3B, we now apply it to T0 (with 11 billion parameters) and compare performance to strong few-shot ICL baselines. From this point onwards, we use exactly the same recipe and hyperparameters across all tasks.

## 4.1 Performance on T0 tasks

First, we evaluate `T-Few` on the datasets that were held out from T0's training mixture. We compare against zero-shot learning with T0 [1] (since we found few-shot ICL to performed worse than zero-shot for T0, see appendix E); few-shot ICL with T5+LM [14] (the next-step-prediction language model upon which T0 is based); and few-shot ICL with the 6.7, 13, and 175 billion parameter variants of GPT-3. See appendix E for more details on these baselines. The accuracy on the held-out T0 datasets (described in section 3.1) is shown in table 1 and fig. 3, with per-dataset results reported in appendix E. We find that `T-Few` outperforms all other methods by a substantial margin. Notably, `T-Few` achieves a 6% higher accuracy than few-shot ICL with GPT-3 175B despite being about $16\times$ smaller and outperforms the smaller GPT-3 variants by an even larger margin. `T-Few` also attains significantly higher accuracy than both zero-shot learning with T0 and few-shot ICL with T5+LM.

| Method | Inference FLOPs | Training FLOPs | Disk space | Acc. |
|---|---|---|---|---|
| `T-Few` | 1.1e12 | 2.7e16 | 4.2 MB | 72.4% |
| T0 [1] | 1.1e12 | 0 | 0 B | 66.9% |
| T5+LM [14] | 4.5e13 | 0 | 16 kB | 49.6% |
| GPT-3 6.7B [4] | 5.4e13 | 0 | 16 kB | 57.2% |
| GPT-3 13B [4] | 1.0e14 | 0 | 16 kB | 60.3% |
| GPT-3 175B [4] | 1.4e15 | 0 | 16 kB | 66.6% |

Table 1: Accuracy on held-out T0 tasks and computational costs for different few-shot learning methods and models. `T-Few` attains the highest accuracy with $1{,}000\times$ lower computational cost than ICL with GPT-3 175B. Fine-tuning with `T-Few` costs about as much as ICL on 20 examples with GPT-3 175B.

| Method | Acc. |
|---|---|
| `T-Few` | 75.8% |
| Human baseline [2] | 73.5% |
| PET [50] | 69.6% |
| SetFit [51] | 66.9% |
| GPT-3 [4] | 62.7% |

Table 2: Top-5 best methods on RAFT as of writing. `T-Few` is the first method to outperform the human baseline and achieves over 6% higher accuracy than the next-best method.

## 4.2 Comparing computational costs

Having established that `T-Few` significantly outperforms ICL-based models, we now compare the relative costs of each few-shot learning approach. For simplicity, we use the FLOPs-per-token estimates for Transformer-based language models introduced by Kaplan et al. [20]. Specifically, we estimate that a decoder-only Transformer (e.g. the GPT series) with $N$ parameters uses $2N$ FLOPs per token for inference and $6N$ FLOPs per token for training. Encoder-decoder models like T0 and T5 (where the encoder and decoder have the same number of layers and layer sizes) only process each token with either the encoder or decoder (each having roughly half the parameters of the full model), so the FLOPs per token estimates are halved to $N$ and $3N$ FLOPs per token for inference and training. We note that FLOPs are not a direct measurement of real-world computational cost because latency, power usage, and other costs can vary significantly depending on hardware and other factors [52]. However, we focus on FLOPs because it is a hardware-independent metric that closely with real-world costs the hardware setup used for running the different methods we consider would likely vary significantly across methods. We summarize the costs in table 1 and discuss them below. For all estimates, we use the median number of shots (41) across the datasets we consider. Rank evaluation and our unlikelihood loss both require processing every possible output choice to attain a prediction for an unlabeled example. The median combined tokenized sequence length for the input and all possible targets is 103 for the datasets we consider. For in-context examples processed for few-shot ICL, only the correct target is required, producing a median sequence length of 98. Assuming that key and value vectors are cached, processing a single example with ICL therefore involves processing $41 \times 98 + 103$ tokens. A summary of our cost estimates is provided in table 1.

**Inference cost.** Beyond improved accuracy, the primary advantage of avoiding few-shot ICL is dramatically lower inference costs. Processing a single input and all target choices with `T-Few`

requires $11\text{e}9 \times 103 = 1.1\text{e}12$ FLOPs, whereas few-shot ICL with GPT-3 175B requires $2 \times 175\text{e}9 \times (41 \times 98 + 103) = 1.4\text{e}15$ FLOPs – more than 3 orders of magnitude more. Inference costs with ICL using the smaller GPT-3 variants are also dramatically higher than the inference cost of `T-Few`. As discussed in section 2.1, caching the key and value vectors of the in-context examples can reduce the computational cost of ICL. However, this would only result in an approximately $41\times$ reduction, which is not nearly enough to make any of the GPT-3 ICL costs as low as `T-Few`.

**Training cost.** Since `T-Few` is the only method that involves updating parameters, it is the only method that incurs a training cost. Training an eleven billion parameter encoder-decoder model for 1,000 steps with a batch size of 8 length-103 sequences requires approximately $3 \times 11\text{e}9 \times 1,000 \times 8 \times 103 = 2.7\text{e}16$ FLOPs. While not insignificant, this is only about 20 times larger than the FLOPs required to process a *single* example with few-shot ICL using GPT-3 175B. In other words, training `T-Few` costs as much as using GPT-3 175B to process 20 examples with few-shot ICL. We also found that fine-tuning T0 with `T-Few` on a single dataset only takes about a half an hour on a single NVIDIA A100 GPU. As of writing, this would cost about \$2 USD using Microsoft Azure.[2]

**Storage cost.** `T-Few` also incurs the largest storage cost. When stored as single-precision floats, the parameters added by $(IA)^3$ take up 4.2 MB of space on disk. In contrast, ICL methods only require storing the tokenized in-context examples (typically stored as 32-bit integers), resulting in a smaller $41 \times 98 \times 32$ bits $= 16$ kB disk space requirement. However, we note that 4.2 MB is dwarfed by the on-disk size of the model checkpoints themselves – storing the $(IA)^3$ adaptation vectors for 10,000 tasks would take about as much space as the T0 checkpoint (41.5 GB).

**Memory usage.** During inference, the primary memory cost is incurred by the model's parameters. The only model smaller than T0 (used by `T-Few`) is GPT-3 6.7B; otherwise, `T-Few` will incur a lower memory cost during inference. Additional memory costs are incurred when training `T-Few` due to the need to cache intermediate activations for backpropagation and for the gradient accumulator variables in Adafactor. However, as mentioned above, a single 80GB A100 GPU is enough for `T-Few`.

### 4.3 Performance on Real-world Few-shot Tasks (RAFT)

So far, we have run evaluatation on a collection of datasets not explicitly designed for benchmarking few-shot learning. To better evaluate `T-Few` in the real world, we took our approach to the RAFT benchmark [2]. RAFT comprise 11 "economically valuable" tasks that mirror real-world applications. Importantly, each RAFT datasets has only 50 training examples with no validation set and a (larger) test set with no public labels, so it is impossible to "cheat" by tuning on an unrealistically-large validation set or by peeking at the test set [32, 31]. We apply `T-Few` to RAFT by using the standard prompts released alongside the dataset. The accuracy of the current top-5 methods is shown in table 2, with further details in appendix G. `T-Few` attains a state-of-the-art accuracy of 75.8% and outperforms the human baseline (73.5% accuracy) for the first time. The next-best model (from Schick and Schütze [50]) achieves 6% lower accuracy and GPT-3 175B attains only 62.7%. These results validate that `T-Few` can be readily applied as-is to novel real-world tasks for strong performance.

### 4.4 Ablation experiments

Given that our `T-Few` design experiments were on T0-3B, we perform an ablation of some of the ingredients of `T-Few` on T0. Results are shown in appendix F. While the gains from adding each ingredient does not always significant increase the accuracy on each individual dataset, each ingredient consistently improves the average performance across datasets: Removing pre-training decreases accuracy by 1.6%, removing unlikelihood training and length normalization decreases accuracy by 4.1%, and removing both pre-training and our additional loss terms reduces accuracy by 2.5%.

## 5   Related Work

Currently, prompt tuning is one of the most parameter-efficient methods for large language models [29, 14, 53]. Liu et al. [54] introduce several tricks to improve prompt tuning, An et al. [55] tune

---

[2]https://docs.microsoft.com/en-us/azure/virtual-machines/ndm-a100-v4-series

prompts along with input embeddings for boost in performance, and Chen et al. [56] improve prompt embeddings through continued pre-training. Given optimization difficulties when training prompt embeddings, Diao et al. [57] recently used black-box optimization to train prompt embeddings without requiring gradients. Several works have analyzed prompt tuning from the perspective of interpretability Khashabi et al. [58] and its similarity to other PEFT methods He et al. [30]. Prompt tuning has been applied to various applications for NLP including continual learning [59], model robustness [60, 61], summarization [62], machine translation [63], co-training [64], probing language models [65, 65], inverse prompting [66] and transfer learning [67]. He et al. [68] recently proposed the use of a hypernetwork to predict prompts for new tasks (rather than training the prompt parameters with gradient descent). Prompt tuning and other PEFT methods have also been explored outside of the context of language models (e.g. vision [22, 69] and vision-and-language models [26]).

Separately, various studies have considered few-shot full-model fine-tuning with discrete prompts [70]. Recent work has analyzed training with discrete prompts, demonstrating a boost in performance with prompting when training on various numbers of examples [71], finding that models perform similarly when trained on good and bad prompts [11], and exploring which prompts work well for few-shot and full-shot setting [72]. There have also been efforts to develop methods that find discrete prompts [73, 74] and training prompts using methods similar to prompt tuning [75].

There has also been a great deal of work on improving ICL. Chen et al. [5], Min et al. [6] use ICL for meta-learning to perform few-shot learning on new tasks. Lampinen et al. [7] show ICL can improve when explanations are provided and [8] use ICL with text retrieved from the web for open-domain question-answering. Meanwhile, Min et al. [9] analyze how ICL works and show that ICL can still perform well when incorrect labels are provided for the in-context examples.

With the advent of billion-parameter language models, there has been a great deal of recent interest in PEFT methods and their compatibility in the few-shot setting. Mahabadi et al. [28] found that PEFT outperforms standard fine-tuning in the low-resource setting. In concurrent work, Mahabadi et al. [76] find that PEFT compares favorably in few-shot fine-tuning against discrete prompts (e.g. PET [70]). Also concurrently, Moosavi et al. [77] propose a framework for introducing adapters whose architecture and design vary from task to task and demonstrate improved results in few-shot settings. Gu et al. [18] and Vu et al. [19] both explored how pre-training prompt tuning parameters can improve when limited labeled data is available. For few-shot learning, Triantafillou et al. [78] explore learning universal and dataset dependent parameters that can be blended for generalization. Requeima et al. [79] use conditional neural adaptive processes and Li et al. [80] leverage distillation from multiple feature extractors to learn new classes or domains in few-shot learning.

# 6 Conclusion

We introduced `T-Few`, a parameter-efficient few-shot learning recipe that attains higher accuracy than few-shot ICL at a lower computational cost. `T-Few` uses $(IA)^3$, a new PEFT method that rescales inner activations with learned vectors. Using $(IA)^3$ produces better performance than fine-tuning the full model while introducing minimal additional parameters. `T-Few` also uses two additional loss terms that encourage the model to output lower probabilities for incorrect choices and account for the length of different answer choices. When applying `T-Few` as-is (with no task-specific hyperparameter tuning or other changes) to the RAFT benchmark, we attained super-human performance for the first time and outperformed prior submissions by a large margin. Through detailed characterization of computational costs, we found that `T-Few` uses over $1,000\times$ fewer FLOPs during inference than few-shot ICL with GPT-3 and only requires 30 minutes to train on a single NVIDIA A100 GPU. Since all of our experiments were on classification tasks, we are interested in applying `T-Few` to generative tasks like as summarization and question answering in future work. We hope our results provide a new perspective on how best to perform few-shot learning with large language models.

## Acknowledgments and Disclosure of Funding

We thank Brian Lester and Noah Constant for helpful discussion on debugging prompt tuning and Rabeeh Karimi Mahabadi for help with Compacter and Intrinsic SAID. We also thank Stella Biderman and the Google TPU Research Cloud who provided valuable computational resources to support this work. This work was supported by NSF-AI Engage Institute DRL-2112635.

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
