## A  Compute resources used

All T0-3B models were trained on 48GB A6000s. Training T0-3B with different PEFT methods took about an hour to train, except for Intrinsic SAID and FishMask which each took about two hours to train. Pre-training $(IA)^3$ took 1 day on 4 A6000s. All T0 models were trained 80GB A100s from DataCrunch [3] and took about half an hour to train each. Pre-training $(IA)^3$ took about 1 day on 4 A100s.

## B  Full Unlikelihood Training and Length Normalization Results

Table 3 shows the full results with unlikelihood training and length normalization.

|          | COPA | H-Swag | StoryCloze | Winogrande | WSC | WiC |
|----------|------|--------|------------|------------|-----|-----|
| FT       | $78.0_{2.0}$ | $39.2_{0.2}$ | $91.5_{1.0}$ | $54.5_{0.9}$ | $66.4_{1.0}$ | $53.8_{1.7}$ |
| + UL     | $81.0_{3.0}$ | $46.1_{4.8}$ | $93.6_{2.5}$ | $56.5_{2.2}$ | $61.5_{8.7}$ | $56.4_{4.1}$ |
| + LN     | $86.0_{4.0}$ | $47.1_{22.4}$ | $94.0_{0.6}$ | $56.9_{3.8}$ | $65.4_{3.9}$ | $53.9_{2.0}$ |
| + UL + LN | $81.0_{11.0}$ | $46.4_{8.8}$ | $93.8_{2.7}$ | $56.5_{1.5}$ | $65.4_{7.7}$ | $57.7_{3.9}$ |

|          | RTE | CB | ANLI-R1 | ANLI-R2 | ANLI-R3 |
|----------|-----|-----|---------|---------|---------|
| FT       | $75.8_{5.4}$ | $82.1_{5.4}$ | $47.8_{1.5}$ | $40.6_{0.8}$ | $37.8_{1.8}$ |
| + UL     | $77.6_{1.4}$ | $89.3_{1.8}$ | $47.9_{1.9}$ | $40.9_{1.9}$ | $38.8_{5.0}$ |
| + LN     | $75.8_{4.3}$ | $89.3_{7.1}$ | $48.2_{0.6}$ | $40.9_{0.9}$ | $38.3_{1.6}$ |
| + UL + LN | $79.8_{3.6}$ | $87.5_{5.4}$ | $46.6_{2.5}$ | $41.3_{0.9}$ | $40.2_{5.3}$ |

Table 3: Per-dataset results for comparing the effect of including the additional loss terms introduced in section 3.2. Subscripts are IQR.

## C  Full PEFT Results

We compare against the following PEFT methods, using a linear decay with warmup scheduler with a warm-up ratio of 0.06 and the Adafactor optimizer [49]. We show the full per-dataset result of all PEFT methods we considered and ablate the losses. Table 4 includes all losses, Table 5 includes $L_{\mathrm{LN}}$, Table 6 includes $L_{\mathrm{UL}}$, and Table 7 does not include either loss.

**Full Model Fine-tuning**  We train for 300 steps with a learning rate of $3e^{-4}$.

**BitFit [47]**  We train for 300 steps with a learning rate of $3e^{-4}$.

**LayerNorm**  We train for 300 steps with a learning rate of $3e^{-4}$.

**Adapter [23]**  We use a reduction factor of 32, ReLU nonlinearity, and residual connections. We train for 500 steps with a learning rate of $3e^{-3}$.

**Compacter [28]**  We train for 500 steps with a learning rate of $3e^{-3}$ and hyper complex division factor of 4 ($n = 4$).

**Compacter++ [28]**  We train for 500 steps with a learning rate of $3e^{-3}$ and hyper complex division factor of 4 ($n = 4$).

**Prompt tuning [14]**  We also add prompt embeddings to the decoder since in prelimary experiments it performed slightly better. We train for 1000 steps with a learning rate of $3e^{-1}$ and use 10 and 100 prompt embeddings.

**Prefix tuning [29]**  We train for 1000 steps with a learning rate of $3e^{-3}$ and adopt the two-layer MLP parameterization in the paper with hidden size 512. We use "Question:" and "Answer:" as initialization text for the prefixes attached to the input and target sequence, respectively.

**FishMask [26]**  The Fisher is first computed on the training examples and we keep $0.2\%$ or $0.02\%$ of the parameters. Then, these parameters are trained for 1500 steps with a learning rate of $3e^{-4}$.

---

[3] https://cloud.datacrunch.io/

**Intrinsic SAID [27]** We train for 3000 steps with a learning rate of $3e^{-2}$. Due to large model size, we use Intrinsic SAID to produce rank-1 updates for 2D weights via an outer product of two vectors.

**LoRA [13]** We use a rank of $4$ with initialization scale of $0.01$ and update all the attention and feedforward module. We train for 1000 steps with a learning rate of $3e^{-3}$.

## D  Full Pre-training Results

Table 8 shows the per-dataset results for of pre-training (IA)³.

## E  Full Main Results

We compare against the following baselines:

**T0.**  To measure the improvement in performance conferred through parameter-efficient few-shot learning, we compare to zero-shot evaluation using T0 itself. In preliminary experiments, we found that T0 was not able to perform few-shot ICL – performance actually *decreased* as we increased the number of in-context examples. This is likely because of the zero-shot format used during multitask prompted fine-tuning and corroborates a recent finding by [10].

**T5+LM.**  Since T0 is unable to perform ICL on its own, we also compare to T5+LM, the next-step-prediction language model upon which T0 is based. Specifically, we use the LM-adapted variant of T5.1.1.xxl released by Lester et al. [14], which has the same architecture and number of parameters as T0. Due to memory constraints and because of its improved performance, we use ensemble ICL for T5+LM [6]. Specifically, we perform one-shot ICL using each example in the training set individually and average the predictions for a given query example. For fair comparison with GPT-3 models, we use the EleutherAI evaluation harness [81], which was designed to replicate the evaluation setup done by Brown et al. [4].

**GPT-3.**  For a strong ICL baseline, we consider models in the GPT-3 family [4]. Specifically, we compare to the 6.7, 13, and 175 billion parameter variants of GPT-3. Because these models have not been publicly released, we report numbers directly from Brown et al. [4]. While GPT-3 is available through the commercial OpenAI API, re-running evaluation through the API would be more than an order of magnitude more expensive than running all of the experiments performed for this paper.

## F  Full Ablation Results

Table table 10 shows the T-Few ablation results.

## G  RAFT Experiment Details

RAFT consists of 11 tasks: Ade Corpus V2, Banking 77, NeurIps Impact Statement Risks, One Stop English, Overruling, Systematic Review Inclusion, Tai Safety Research, Terms of Service, Tweet Eval Hate, and Twitter Complaints. We use the T-Few recipe on all datasets without putting the labels into the input string except Banking 77. Since Banking 77 has 77 classes which causes memory issues for unlikelihood training, we turn off unlikelihood training for Banking 77. We also feed in all the labels as part of the input string for Banking 77 since there were some labels never seen during training and clean the labels by replacing "." with ",".

Per-dataset results of T-Few and the other top-5 methods on RAFT are shown in table 11.

| | # of Param | COPA | H-Swag | StoryCloze | Winogrande |
|---|---|---|---|---|---|
| Full Model Fine-tuning | 3B | $81.0_{11.0}$ | $46.4_{8.8}$ | $93.8_{2.7}$ | $56.5_{1.5}$ |
| BitFit (with LayerNorm) | 1.3M | $75.0_{2.0}$ | $29.5_{3.6}$ | $88.6_{0.7}$ | $49.6_{1.3}$ |
| LayerNorm | 250K | $76.0_{2.0}$ | $29.6_{3.4}$ | $88.7_{0.9}$ | $49.4_{1.4}$ |
| Adapter | 12.9M | $84.0_{3.0}$ | $41.9_{3.8}$ | $91.7_{3.7}$ | $54.7_{3.6}$ |
| Compacter | 807K | $84.0_{5.0}$ | $46.4_{2.5}$ | $93.5_{2.2}$ | $55.5_{2.9}$ |
| Compacter++ | 540K | $86.0_{3.0}$ | $46.3_{3.0}$ | $93.5_{1.2}$ | $55.1_{1.1}$ |
| Prompt tuning (10) | 41K | $67.0_{5.0}$ | $29.9_{0.6}$ | $84.2_{0.8}$ | $51.9_{1.6}$ |
| Prompt tuning (100) | 409K | $60.0_{19.0}$ | $26.8_{0.6}$ | $74.0_{3.4}$ | $51.1_{0.8}$ |
| Prefix tuning | 576K | $71.0_{8.0}$ | $42.1_{4.0}$ | $90.2_{3.1}$ | $52.0_{1.3}$ |
| FishMask (0.2%) | 6M | $82.0_{5.0}$ | $44.1_{4.2}$ | $94.2_{1.8}$ | $54.5_{2.1}$ |
| FishMask (0.02%) | 600K | $84.0_{6.0}$ | $38.2_{3.6}$ | $93.6_{0.7}$ | $53.9_{2.8}$ |
| Intrinsic SAID (20K) | 20K | $76.0_{4.0}$ | $38.3_{6.4}$ | $89.7_{2.7}$ | $50.9_{1.0}$ |
| Intrinsic SAID (500K) | 500K | $77.0_{4.0}$ | $36.7_{4.5}$ | $89.3_{2.3}$ | $52.7_{2.1}$ |
| LoRA | 9.1M | $88.0_{5.0}$ | $47.1_{3.2}$ | $93.6_{2.1}$ | $56.8_{3.3}$ |
| $(IA)^3$ | 540K | $87.0_{3.0}$ | $49.4_{4.6}$ | $94.7_{2.7}$ | $59.8_{0.6}$ |

| | # of Param | WSC | WiC | RTE | CB |
|---|---|---|---|---|---|
| Full Model Fine-tuning | 3B | $65.4_{7.7}$ | $57.7_{3.9}$ | $79.8_{3.6}$ | $87.5_{5.4}$ |
| BitFit (with LayerNorm) | 1.3M | $61.5_{11.5}$ | $51.7_{2.2}$ | $72.2_{1.1}$ | $57.1_{1.8}$ |
| LayerNorm | 250K | $63.5_{12.5}$ | $52.2_{1.6}$ | $71.8_{0.4}$ | $57.1_{1.8}$ |
| Adapter | 12.9M | $65.4_{1.0}$ | $55.5_{2.7}$ | $76.2_{3.6}$ | $87.5_{3.6}$ |
| Compacter ($n = 4$) | 807K | $64.4_{6.7}$ | $55.2_{3.8}$ | $75.8_{6.1}$ | $82.1_{3.6}$ |
| Compacter++ ($n = 4$) | 540K | $65.4_{3.9}$ | $54.1_{2.2}$ | $76.9_{0.4}$ | $82.1_{3.6}$ |
| Prompt tuning (10) | 41K | $54.8_{10.6}$ | $51.6_{2.0}$ | $52.7_{5.4}$ | $66.1_{1.8}$ |
| Prompt tuning (100) | 409K | $60.6_{4.8}$ | $50.0_{1.1}$ | $48.0_{2.9}$ | $53.6_{17.9}$ |
| Prefix tuning | 576K | $56.7_{3.3}$ | $54.2_{3.3}$ | $68.6_{3.3}$ | $84.0_{1.8}$ |
| FishMask (0.2%) | 6M | $63.5_{4.8}$ | $52.5_{3.3}$ | $76.9_{4.7}$ | $83.9_{3.6}$ |
| FishMask (0.02%) | 600K | $61.5_{1.0}$ | $53.5_{1.3}$ | $75.5_{5.4}$ | $76.8_{3.6}$ |
| Intrinsic SAID (20K) | 20K | $55.8_{6.7}$ | $55.3_{0.5}$ | $66.1_{5.4}$ | $83.9_{1.8}$ |
| Intrinsic SAID (500K) | 500K | $61.5_{8.7}$ | $55.0_{2.7}$ | $69.0_{7.6}$ | $80.4_{0.0}$ |
| LoRA | 9.1M | $60.6_{5.8}$ | $55.2_{5.0}$ | $78.3_{7.6}$ | $85.7_{1.8}$ |
| $(IA)^3$ | 540K | $68.3_{6.7}$ | $56.0_{4.6}$ | $78.0_{2.5}$ | $87.5_{1.8}$ |

| | # of Param | ANLI-R1 | ANLI-R2 | ANLI-R3 |
|---|---|---|---|---|
| Full Model Fine-tuning | 3B | $46.6_{2.5}$ | $41.3_{0.9}$ | $40.2_{5.3}$ |
| BitFit (with LayerNorm) | 1.3M | $36.5_{0.8}$ | $35.3_{2.2}$ | $36.6_{0.8}$ |
| LayerNorm | 250K | $36.5_{0.7}$ | $35.1_{2.6}$ | $36.3_{1.0}$ |
| Adapter | 12.9M | $45.1_{2.6}$ | $40.4_{1.2}$ | $35.3_{1.3}$ |
| Compacter | 807K | $40.8_{3.3}$ | $37.4_{0.2}$ | $35.8_{3.3}$ |
| Compacter++ | 540K | $41.7_{0.4}$ | $38.3_{1.8}$ | $36.9_{1.5}$ |
| Prompt tuning (10) | 41K | $34.2_{1.9}$ | $33.5_{1.1}$ | $33.5_{1.3}$ |
| Prompt tuning (100) | 409K | $33.4_{1.2}$ | $33.8_{0.5}$ | $33.3_{0.8}$ |
| Prefix tuning | 576K | $43.3_{4.1}$ | $37.5_{1.2}$ | $36.5_{1.5}$ |
| FishMask (0.2%) | 6M | $43.7_{0.3}$ | $39.7_{1.4}$ | $37.2_{1.1}$ |
| FishMask (0.02%) | 600K | $39.9_{0.9}$ | $38.1_{2.0}$ | $36.2_{1.8}$ |
| Intrinsic SAID (20K) | 20K | $41.3_{1.3}$ | $38.5_{1.8}$ | $35.8_{2.0}$ |
| Intrinsic SAID (500K) | 500K | $40.4_{3.3}$ | $35.4_{4.1}$ | $35.5_{1.6}$ |
| LoRA | 9.1M | $45.1_{2.5}$ | $41.0_{1.4}$ | $39.5_{4.8}$ |
| $(IA)^3$ | 540K | $48.6_{2.0}$ | $40.8_{1.5}$ | $40.8_{2.3}$ |

Table 4: Per-dataset accuracies for the PEFT methods we consider when adding $L_{\text{UL}}$ and $L_{\text{LN}}$. Subscripts are IQR.

| | # of Param | COPA | H-Swag | StoryCloze | Winogrande |
|---|---|---|---|---|---|
| Full Model Fine-tuning | 3B | $86.0_{4.0}$ | $47.1_{22.4}$ | $93.9_{0.5}$ | $56.9_{3.7}$ |
| BitFit (with LayerNorm) | 1.3M | $80.0_{6.0}$ | $31.3_{0.1}$ | $92.8_{0.2}$ | $51.3_{0.7}$ |
| LayerNorm | 250K | $82.0_{2.0}$ | $31.2_{0.6}$ | $92.8_{0.4}$ | $51.1_{0.3}$ |
| Adapter | 12.9M | $84.0_{5.0}$ | $44.0_{3.2}$ | $92.8_{2.3}$ | $52.6_{0.5}$ |
| Compacter ($n = 4$) | 807K | $85.0_{3.0}$ | $47.2_{5.3}$ | $94.3_{1.2}$ | $53.9_{1.3}$ |
| Compacter++ ($n = 4$) | 540K | $85.0_{2.0}$ | $47.8_{1.6}$ | $94.5_{0.6}$ | $54.3_{2.9}$ |
| Prompt tuning (10) | 41K | $72.0_{5.0}$ | $30.4_{1.0}$ | $90.3_{1.2}$ | $50.5_{0.9}$ |
| Prompt tuning (100) | 409K | $65.0_{1.0}$ | $27.9_{4.6}$ | $87.0_{3.0}$ | $51.9_{0.3}$ |
| Prefix tuning | 576K | $79.0_{6.0}$ | $34.4_{9.7}$ | $90.3_{3.1}$ | $51.1_{1.7}$ |
| FishMask (0.2%) | 6M | $85.0_{4.0}$ | $43.3_{3.1}$ | $93.8_{0.9}$ | $54.3_{0.1}$ |
| FishMask (0.0%) | 600K | $82.0_{2.0}$ | $31.2_{1.3}$ | $93.6_{1.1}$ | $53.9_{1.9}$ |
| Intrinsic SAID (20K) | 20K | $67.0_{8.0}$ | $28.9_{0.7}$ | $90.3_{0.3}$ | $52.2_{1.9}$ |
| Intrinsic SAID (500K) | 500K | $63.0_{1.0}$ | $27.6_{1.2}$ | $79.2_{3.8}$ | $51.2_{2.3}$ |
| LoRA | 9.1M | $86.0_{1.0}$ | $48.6_{2.6}$ | $94.4_{1.6}$ | $56.1_{1.0}$ |
| (IA)$^3$ | 540K | $90.0_{2.0}$ | $50.0_{3.0}$ | $95.4_{1.1}$ | $58.2_{0.5}$ |

| | # of Param | WSC | WiC | RTE | CB |
|---|---|---|---|---|---|
| Full Model Fine-tuning | 3B | $65.3_{3.8}$ | $53.9_{2.0}$ | $75.8_{4.3}$ | $89.2_{7.1}$ |
| BitFit (with LayerNorm) | 1.3M | $63.4_{2.8}$ | $54.2_{3.1}$ | $75.4_{1.8}$ | $67.8_{0.0}$ |
| LayerNorm | 250K | $60.5_{2.8}$ | $55.3_{1.8}$ | $76.1_{1.4}$ | $67.8_{1.7}$ |
| Adapter | 12.9M | $63.4_{3.8}$ | $55.4_{3.6}$ | $77.2_{3.9}$ | $80.3_{3.5}$ |
| Compacter ($n = 4$) | 807K | $64.4_{3.8}$ | $53.2_{5.4}$ | $75.4_{2.8}$ | $82.1_{5.3}$ |
| Compacter++ ($n = 4$) | 540K | $65.3_{3.8}$ | $54.8_{3.4}$ | $77.2_{5.7}$ | $76.7_{7.1}$ |
| Prompt tuning (10) | 41K | $53.8_{4.8}$ | $52.0_{1.7}$ | $55.2_{2.5}$ | $66.0_{3.5}$ |
| Prompt tuning (100) | 409K | $50.9_{6.7}$ | $51.8_{1.5}$ | $48.3_{3.6}$ | $62.5_{12.5}$ |
| Prefix tuning | 576K | $60.5_{3.8}$ | $68.9_{0.7}$ | $80.3_{12.5}$ | $75.0_{8.9}$ |
| FishMask (0.2%) | 6M | $66.3_{2.8}$ | $54.2_{1.1}$ | $75.8_{3.6}$ | $83.9_{7.1}$ |
| FishMask (0.0%) | 600K | $60.5_{1.9}$ | $52.8_{1.1}$ | $75.0_{3.6}$ | $76.7_{3.5}$ |
| Intrinsic SAID (20K) | 20K | $57.6_{6.7}$ | $54.0_{4.3}$ | $68.9_{1.4}$ | $80.3_{1.7}$ |
| Intrinsic SAID (500K) | 500K | $60.5_{13.4}$ | $54.8_{0.9}$ | $69.6_{1.4}$ | $82.1_{5.3}$ |
| LoRA | 9.1M | $61.5_{1.9}$ | $55.0_{4.7}$ | $74.7_{4.6}$ | $85.7_{1.7}$ |
| (IA)$^3$ | 540K | $66.3_{3.8}$ | $53.7_{0.6}$ | $76.9_{2.8}$ | $83.9_{0.0}$ |

| | # of Param | ANLI-R1 | ANLI-R2 | ANLI-R3 | **Avg.** |
|---|---|---|---|---|---|
| Full Model Fine-tuning | 3B | $48.2_{0.6}$ | $40.9_{0.9}$ | $38.2_{1.5}$ | 63.2 |
| BitFit (with LayerNorm) | 1.3M | $36.1_{1.4}$ | $35.6_{1.4}$ | $35.4_{2.0}$ | 56.7 |
| LayerNorm | 250K | $37.3_{0.5}$ | $37.1_{0.7}$ | $36.2_{1.0}$ | 57.0 |
| Adapter | 12.9M | $42.4_{3.2}$ | $38.8_{0.6}$ | $36.5_{3.8}$ | 60.7 |
| Compacter ($n = 4$) | 807K | $42.9_{3.9}$ | $38.0_{0.8}$ | $37.3_{2.3}$ | 61.2 |
| Compacter++ ($n = 4$) | 540K | $41.9_{0.5}$ | $38.5_{2.4}$ | $36.0_{0.5}$ | 61.1 |
| Prompt tuning (10) | 41K | $34.2_{1.1}$ | $34.2_{1.3}$ | $34.4_{0.8}$ | 52.1 |
| Prompt tuning (100) | 409K | $34.1_{1.1}$ | $34.2_{0.2}$ | $34.0_{1.2}$ | 49.8 |
| Prefix tuning | 576K | $37.5_{3.6}$ | $34.1_{4.5}$ | $34.4_{9.7}$ | 58.7 |
| FishMask (0.2%) | 6M | $43.4_{0.6}$ | $40.0_{0.9}$ | $36.7_{2.8}$ | 60.0 |
| FishMask (0.02%) | 600K | $40.1_{0.9}$ | $38.0_{2.0}$ | $35.5_{0.7}$ | 57.7 |
| Intrinsic SAID (20K) | 20K | $38.8_{2.0}$ | $37.4_{2.0}$ | $34.1_{2.3}$ | 55.4 |
| Intrinsic SAID (500K) | 500K | $40.5_{3.2}$ | $36.8_{1.9}$ | $34.5_{1.5}$ | 54.5 |
| LoRA | 9.1M | $46.2_{1.7}$ | $41.4_{0.9}$ | $38.4_{2.6}$ | 62.5 |
| (IA)$^3$ | 540K | $49.2_{2.8}$ | $40.3_{2.3}$ | $40.4_{3.1}$ | 64.0 |

Table 5: Per-dataset accuracies for the PEFT methods we consider when adding $L_{\mathrm{LN}}$. Subscripts are IQR.

| | # of Param | COPA | H-Swag | StoryCloze | Winogrande |
|---|---|---|---|---|---|
| Full Model Fine-tuning | 3B | $81.0_{3.0}$ | $46.1_{4.8}$ | $93.6_{2.5}$ | $56.5_{2.2}$ |
| BitFit (with LayerNorm) | 1.3M | $81.0_{4.0}$ | $35.5_{2.3}$ | $92.7_{0.8}$ | $50.9_{0.0}$ |
| LayerNorm | 250K | $82.0_{1.0}$ | $34.6_{2.3}$ | $92.6_{0.7}$ | $51.7_{1.2}$ |
| Adapter | 12.9M | $83.0_{1.0}$ | $42.5_{5.3}$ | $90.4_{3.1}$ | $53.6_{3.6}$ |
| Compacter ($n=4$) | 807K | $88.0_{3.0}$ | $42.9_{4.0}$ | $92.8_{1.8}$ | $54.6_{1.5}$ |
| Compacter++ ($n=4$) | 540K | $85.0_{2.0}$ | $48.2_{2.9}$ | $93.8_{1.6}$ | $54.8_{2.8}$ |
| Prompt tuning (10) | 41K | $74.0_{5.0}$ | $29.2_{2.4}$ | $88.8_{1.1}$ | $51.3_{0.4}$ |
| Prompt tuning (100) | 409K | $68.0_{7.0}$ | $28.5_{2.4}$ | $86.9_{4.3}$ | $50.5_{0.1}$ |
| Prefix tuning | 576K | $69.0_{2.0}$ | $29.0_{10.8}$ | $86.4_{2.3}$ | $50.6_{1.4}$ |
| FishMask (0.2%) | 6M | $85.0_{5.0}$ | $42.5_{3.4}$ | $94.0_{1.5}$ | $53.6_{2.6}$ |
| FishMask (0.0%) | 600K | $84.0_{4.0}$ | $38.4_{3.1}$ | $93.1_{1.2}$ | $53.5_{2.2}$ |
| Intrinsic SAID (20K) | 20K | $74.0_{3.0}$ | $38.7_{5.1}$ | $89.7_{1.6}$ | $51.7_{1.9}$ |
| Intrinsic SAID (500K) | 500K | $76.0_{7.0}$ | $37.9_{4.3}$ | $89.2_{2.1}$ | $50.9_{0.6}$ |
| LoRA | 9.1M | $87.0_{3.0}$ | $46.9_{1.9}$ | $93.1_{2.0}$ | $57.9_{3.6}$ |
| (IA)$^3$ | 540K | $86.0_{4.0}$ | $48.7_{4.1}$ | $94.0_{2.8}$ | $58.7_{1.3}$ |

| | # of Param | WSC | WiC | RTE | CB |
|---|---|---|---|---|---|
| Full Model Fine-tuning | 3B | $61.5_{8.6}$ | $56.4_{4.0}$ | $77.6_{1.4}$ | $89.2_{1.7}$ |
| BitFit (with LayerNorm) | 1.3M | $64.4_{3.8}$ | $53.6_{2.5}$ | $76.1_{3.6}$ | $60.7_{1.7}$ |
| LayerNorm | 250K | $60.5_{8.6}$ | $53.9_{2.3}$ | $75.0_{1.8}$ | $57.1_{3.5}$ |
| Adapter | 12.9M | $65.3_{6.7}$ | $54.3_{3.1}$ | $79.0_{5.4}$ | $85.7_{3.5}$ |
| Compacter ($n=4$) | 807K | $65.3_{4.8}$ | $54.5_{3.6}$ | $75.4_{5.0}$ | $82.1_{0.0}$ |
| Compacter++ ($n=4$) | 540K | $64.4_{3.8}$ | $55.6_{3.6}$ | $77.6_{4.6}$ | $80.3_{7.1}$ |
| Prompt tuning (10) | 41K | $54.8_{6.7}$ | $52.8_{3.2}$ | $52.7_{1.0}$ | $69.6_{5.3}$ |
| Prompt tuning (100) | 409K | $50.0_{3.8}$ | $50.1_{0.9}$ | $52.7_{4.3}$ | $58.9_{12.5}$ |
| Prefix tuning | 576K | $55.7_{1.9}$ | $71.1_{6.1}$ | $82.1_{5.3}$ | $83.9_{8.9}$ |
| FishMask (0.2%) | 6M | $62.5_{3.8}$ | $53.6_{1.4}$ | $76.1_{2.1}$ | $83.9_{8.9}$ |
| FishMask (0.02%) | 600K | $59.6_{1.9}$ | $53.6_{0.4}$ | $74.3_{5.0}$ | $75.0_{1.7}$ |
| Intrinsic SAID (20K) | 20K | $54.8_{7.6}$ | $55.8_{0.3}$ | $65.3_{9.3}$ | $83.9_{3.5}$ |
| Intrinsic SAID (500K) | 500K | $56.7_{3.8}$ | $55.9_{1.5}$ | $64.6_{9.7}$ | $80.3_{5.3}$ |
| LoRA | 9.1M | $59.6_{12.5}$ | $55.4_{4.8}$ | $79.0_{1.8}$ | $87.5_{1.7}$ |
| (IA)$^3$ | 540K | $65.3_{4.8}$ | $56.7_{4.3}$ | $77.2_{2.5}$ | $87.5_{1.7}$ |

| | # of Param | ANLI-R1 | ANLI-R2 | ANLI-R3 | **Avg.** |
|---|---|---|---|---|---|
| Full Model Fine-tuning | 3B | $47.9_{1.9}$ | $40.9_{1.9}$ | $38.8_{5.0}$ | 62.7 |
| BitFit (with LayerNorm) | 1.3M | $36.4_{1.1}$ | $34.0_{0.7}$ | $35.2_{2.4}$ | 56.4 |
| LayerNorm | 250K | $37.0_{1.9}$ | $36.0_{2.1}$ | $35.5_{2.1}$ | 56.0 |
| Adapter | 12.9M | $43.9_{1.1}$ | $38.6_{1.1}$ | $36.1_{2.1}$ | 61.1 |
| Compacter ($n=4$) | 807K | $41.8_{1.3}$ | $37.6_{3.0}$ | $37.1_{1.9}$ | 61.1 |
| Compacter++ ($n=4$) | 540K | $41.7_{0.6}$ | $38.2_{2.5}$ | $35.5_{0.3}$ | 61.4 |
| Prompt tuning (10) | 41K | $35.0_{2.1}$ | $33.8_{0.6}$ | $33.6_{2.7}$ | 52.3 |
| Prompt tuning (100) | 409K | $35.7_{0.9}$ | $33.8_{1.5}$ | $33.0_{2.1}$ | 49.8 |
| Prefix tuning | 576K | $34.6_{1.6}$ | $36.8_{4.6}$ | $38.5_{3.0}$ | 58.0 |
| FishMask (0.2%) | 6M | $44.1_{1.0}$ | $38.7_{1.5}$ | $38.2_{0.8}$ | 59.7 |
| FishMask (0.02%) | 600K | $40.5_{2.6}$ | $37.0_{1.2}$ | $35.5_{0.7}$ | 57.6 |
| Intrinsic SAID (20K) | 20K | $39.6_{4.2}$ | $36.9_{1.4}$ | $35.5_{0.9}$ | 56.9 |
| Intrinsic SAID (500K) | 500K | $40.2_{1.9}$ | $36.5_{2.1}$ | $34.5_{0.8}$ | 56.6 |
| LoRA | 9.1M | $45.9_{2.2}$ | $41.1_{1.7}$ | $38.8_{1.0}$ | 62.9 |
| (IA)$^3$ | 540K | $49.8_{2.1}$ | $40.3_{0.3}$ | $40.1_{3.3}$ | 64.0 |

Table 6: Per-dataset accuracies for the PEFT methods we consider when adding $L_{\mathrm{UL}}$. Subscripts are IQR.

| | # of Param | COPA | H-Swag | StoryCloze | Winogrande |
|---|---|---|---|---|---|
| Full Model Fine-tuning | 3B | $78.0_{2.0}$ | $39.1_{0.2}$ | $91.4_{0.9}$ | $54.4_{0.8}$ |
| BitFit (with LayerNorm) | 1.3M | $77.0_{7.0}$ | $33.7_{0.3}$ | $90.4_{0.2}$ | $51.5_{0.1}$ |
| LayerNorm | 250K | $77.0_{7.0}$ | $33.5_{0.6}$ | $90.4_{0.2}$ | $51.3_{0.3}$ |
| Adapter | 12.9M | $76.0_{5.0}$ | $36.4_{2.2}$ | $90.5_{1.7}$ | $52.0_{0.4}$ |
| Compacter ($n = 4$) | 807K | $81.0_{5.0}$ | $37.5_{0.6}$ | $91.5_{0.2}$ | $52.5_{0.8}$ |
| Compacter++ ($n = 4$) | 540K | $78.0_{2.0}$ | $37.0_{1.0}$ | $91.9_{0.9}$ | $53.1_{0.8}$ |
| Prompt tuning (10) | 41K | $73.0_{4.0}$ | $30.0_{1.6}$ | $88.8_{1.1}$ | $52.2_{0.3}$ |
| Prompt tuning (100) | 409K | $66.0_{4.0}$ | $26.3_{4.4}$ | $87.4_{0.2}$ | $51.1_{0.5}$ |
| Prefix tuning | 576K | $70.0_{3.0}$ | $27.9_{6.6}$ | $86.7_{2.2}$ | $51.0_{1.1}$ |
| FishMask (0.2%) | 6M | $77.0_{3.0}$ | $35.4_{0.8}$ | $90.5_{1.0}$ | $52.9_{0.8}$ |
| FishMask (0.02%) | 600K | $74.0_{2.0}$ | $31.1_{1.3}$ | $89.5_{1.2}$ | $52.5_{0.4}$ |
| Intrinsic SAID (20K) | 20K | $71.0_{8.0}$ | $30.1_{1.0}$ | $87.8_{2.1}$ | $51.4_{1.9}$ |
| Intrinsic SAID (500K) | 500K | $71.0_{1.0}$ | $28.1_{1.4}$ | $86.4_{1.9}$ | $51.1_{1.6}$ |
| LoRA | 9.1M | $80.0_{5.0}$ | $39.1_{1.2}$ | $92.0_{1.0}$ | $53.7_{0.4}$ |
| $(IA)^3$ | 540K | $82.0_{1.0}$ | $40.5_{0.5}$ | $92.5_{0.4}$ | $56.9_{2.5}$ |

| | # of Param | WSC | WiC | RTE | CB |
|---|---|---|---|---|---|
| Full Model Fine-tuning | 3B | $66.3_{0.9}$ | $53.7_{1.7}$ | $75.8_{5.4}$ | $82.1_{5.3}$ |
| BitFit (with LayerNorm) | 1.3M | $61.5_{3.8}$ | $53.1_{1.7}$ | $76.5_{1.0}$ | $64.2_{8.9}$ |
| LayerNorm | 250K | $61.5_{3.8}$ | $53.2_{1.7}$ | $76.1_{2.1}$ | $62.5_{8.9}$ |
| Adapter | 12.9M | $65.3_{7.6}$ | $54.7_{1.7}$ | $77.2_{2.8}$ | $83.9_{1.7}$ |
| Compacter ($n = 4$) | 807K | $61.5_{2.8}$ | $55.3_{3.6}$ | $76.1_{2.1}$ | $83.9_{0.0}$ |
| Compacter++ ($n = 4$) | 540K | $61.5_{1.9}$ | $54.7_{4.2}$ | $73.6_{1.8}$ | $78.5_{5.3}$ |
| Prompt tuning (10) | 41K | $53.8_{7.6}$ | $52.5_{1.8}$ | $57.4_{4.3}$ | $69.6_{10.7}$ |
| Prompt tuning (100) | 409K | $56.7_{6.7}$ | $52.3_{0.6}$ | $54.1_{3.9}$ | $53.5_{19.6}$ |
| Prefix tuning | 576K | $52.8_{7.6}$ | $52.5_{0.3}$ | $72.5_{11.9}$ | $75.0_{17.8}$ |
| FishMask (0.2%) | 6M | $62.5_{4.8}$ | $54.2_{2.0}$ | $77.2_{5.4}$ | $82.1_{1.7}$ |
| FishMask (0.02%) | 600K | $58.6_{2.8}$ | $54.3_{1.1}$ | $76.1_{5.0}$ | $75.0_{3.5}$ |
| Intrinsic SAID (20K) | 20K | $60.5_{1.9}$ | $56.1_{2.3}$ | $70.4_{4.3}$ | $76.7_{8.9}$ |
| Intrinsic SAID (500K) | 500K | $57.6_{5.7}$ | $55.1_{3.9}$ | $72.9_{4.3}$ | $80.3_{0.0}$ |
| LoRA | 9.1M | $64.4_{12.5}$ | $54.8_{3.4}$ | $77.2_{4.3}$ | $87.5_{3.5}$ |
| $(IA)^3$ | 540K | $64.4_{3.8}$ | $54.2_{1.5}$ | $77.9_{1.8}$ | $82.1_{5.3}$ |

| | # of Param | ANLI-R1 | ANLI-R2 | ANLI-R3 | **Avg.** |
|---|---|---|---|---|---|
| Full Model Fine-tuning | 3B | $47.8_{1.5}$ | $40.6_{0.8}$ | $37.7_{1.8}$ | 60.6 |
| BitFit (with LayerNorm) | 1.3M | $37.3_{1.8}$ | $36.1_{2.6}$ | $35.1_{3.6}$ | 56.0 |
| LayerNorm | 250K | $37.5_{1.5}$ | $36.0_{2.8}$ | $35.0_{3.4}$ | 55.8 |
| Adapter | 12.9M | $40.7_{3.7}$ | $39.2_{1.1}$ | $35.8_{1.9}$ | 59.2 |
| Compacter ($n = 4$) | 807K | $41.8_{2.7}$ | $38.0_{0.8}$ | $36.0_{2.7}$ | 59.5 |
| Compacter++ ($n = 4$) | 540K | $41.1_{1.5}$ | $38.9_{2.5}$ | $36.9_{1.4}$ | 58.6 |
| Prompt tuning (10) | 41K | $33.6_{0.7}$ | $33.8_{1.1}$ | $34.8_{1.0}$ | 52.7 |
| Prompt tuning (100) | 409K | $35.6_{1.7}$ | $34.5_{0.7}$ | $34.7_{1.4}$ | 50.2 |
| Prefix tuning | 576K | $37.6_{2.3}$ | $34.1_{3.5}$ | $35.0_{0.6}$ | 54.1 |
| FishMask (0.2%) | 6M | $43.5_{0.3}$ | $40.3_{0.4}$ | $36.4_{2.2}$ | 59.3 |
| FishMask (0.02%) | 600K | $40.4_{2.2}$ | $37.5_{1.0}$ | $36.4_{1.0}$ | 56.8 |
| Intrinsic SAID (20K) | 20K | $38.9_{2.5}$ | $38.0_{2.0}$ | $34.9_{1.0}$ | 56.0 |
| Intrinsic SAID (500K) | 500K | $38.3_{0.6}$ | $35.8_{1.5}$ | $34.5_{1.0}$ | 55.6 |
| LoRA | 9.1M | $44.2_{2.6}$ | $40.4_{1.2}$ | $37.5_{0.5}$ | 61.0 |
| $(IA)^3$ | 540K | $48.5_{0.9}$ | $40.2_{1.8}$ | $39.4_{1.7}$ | 61.7 |

Table 7: Per-dataset accuracies for the PEFT methods we consider without $L_{\mathrm{UL}}$ or $L_{\mathrm{LN}}$. Subscripts are IQR.

| | COPA | H-Swag | StoryCloze | Winogrande | WSC | WiC |
|---|---|---|---|---|---|---|
| (IA)$^3$ | $87.0_{3.0}$ | $49.4_{4.6}$ | $94.7_{2.7}$ | $59.8_{0.6}$ | $68.3_{6.7}$ | $56.0_{4.6}$ |
| + PT | $89.0_{5.0}$ | $51.2_{4.6}$ | $95.1_{2.5}$ | $62.6_{1.1}$ | $70.2_{8.7}$ | $57.2_{2.5}$ |
| | RTE | CB | ANLI-R1 | ANLI-R2 | ANLI-R3 | **Acc.** |
| (IA)$^3$ | $78.0_{2.5}$ | $87.5_{1.8}$ | $48.6_{2.0}$ | $40.8_{1.5}$ | $40.83_{2.3}$ | 64.6 |
| + PT | $80.9_{1.4}$ | $87.5_{1.8}$ | $49.3_{1.1}$ | $41.1_{0.5}$ | $39.8_{4.8}$ | 65.8 |

Table 8: Per-dataset results when pre-training (PT) (IA)$^3$ vs. not pre-training (IA)$^3$. Subscripts are IQR.

| | COPA | H-Swag | StoryCloze | Winogrande | WSC | WiC |
|---|---|---|---|---|---|---|
| T-Few | $93.0_{2.0}$ | $67.1_{6.0}$ | $97.9_{0.3}$ | $74.3_{1.5}$ | $75.0_{5.5}$ | $62.2_{7.8}$ |
| T0 | 90.8 | 33.7 | 94.7 | 60.5 | 64.4 | 57.2 |
| T5+LM | 68.0 | 60.95 | 62.8 | 56.9 | 63.5 | 50.0 |
| GPT-3 (175B) | 92.0 | 79.3 | 87.7 | 77.7 | 75.0 | 55.3 |
| GPT-3 (13B) | 86.0 | 71.3 | 83.0 | 70.0 | 75.0 | 51.1 |
| GPT-3 (6.7B) | 83.0 | 67.3 | 81.2 | 67.4 | 67.3 | 53.1 |
| | RTE | CB | ANLI-R1 | ANLI-R2 | ANLI-R3 | |
| T-Few | $85.6_{2.9}$ | $87.5_{3.6}$ | $59.3_{3.6}$ | $49.8_{2.6}$ | $44.8_{8.0}$ | |
| T0 | 81.2 | 78.6 | 44.7 | 39.4 | 42.4 | |
| T5 + LM | 53.4 | 32.1 | 33.3 | 32.7 | 34.1 | |
| GPT-3 (175B) | 72.9 | 82.1 | 36.8 | 34.0 | 40.2 | |
| GPT-3 (13B) | 60.6 | 66.1 | 33.3 | 32.6 | 34.5 | |
| GPT-3 (6.7B) | 49.5 | 60.7 | 33.1 | 33.1 | 33.9 | |

Table 9: Comparing T-Few with few-shot ICL methods. All GPT-3 numbers are from Brown et al. [4] and all T0 numbers are from Sanh et al. [1]. Subscripts are IQR.

| | COPA | H-Swag | StoryCloze | Winogrande | WSC | WiC |
|---|---|---|---|---|---|---|
| T-Few | $93.0_{2.0}$ | $67.1_{6.0}$ | $97.9_{0.3}$ | $74.3_{1.5}$ | $75.0_{5.5}$ | $62.15_{7.8}$ |
| - PT | $92.0_{2.0}$ | $64.5_{6.6}$ | $97.8_{0.8}$ | $72.7_{1.0}$ | $73.1_{6.3}$ | $60.8_{6.4}$ |
| - $L_{\mathrm{UL}}$ - $L_{\mathrm{LN}}$ | $91.0_{2.0}$ | $52.1_{2.7}$ | $97.4_{0.5}$ | $71.9_{1.1}$ | $71.2_{1.0}$ | $62.2_{2.4}$ |
| - PT - $L_{\mathrm{UL}}$ - $L_{\mathrm{LN}}$ | $94.0_{2.3}$ | $52.7_{4.9}$ | $98.0_{0.3}$ | $74.0_{1.1}$ | $72.6_{4.8}$ | $62.6_{5.0}$ |
| | RTE | CB | ANLI-R1 | ANLI-R2 | ANLI-R3 | **Acc.** |
| T-Few | $85.6_{2.9}$ | $87.5_{3.6}$ | $59.3_{3.6}$ | $49.8_{2.6}$ | $44.8_{8.0}$ | 72.4 |
| - PT | $84.5_{2.8}$ | $83.9_{5.4}$ | $57.9_{3.2}$ | $48.6_{3.0}$ | $43.1_{5.7}$ | 70.8 |
| - $L_{\mathrm{UL}}$ - $L_{\mathrm{LN}}$ | $82.0_{0.7}$ | $82.1_{3.6}$ | $54.8_{0.4}$ | $46.1_{0.6}$ | $40.8_{5.2}$ | 68.3 |
| - PT - $L_{\mathrm{UL}}$ - $L_{\mathrm{LN}}$ | $84.5_{2.9}$ | $80.4_{3.6}$ | $57.1_{3.1}$ | $47.1_{2.4}$ | $43.8_{5.9}$ | 69.7 |

Table 10: T-Few ablation results when omitting (IA)$^3$ pre-training (PT) and/or the $L_{\mathrm{UL}}$ and $L_{\mathrm{LN}}$ losses. Subscripts are IQR.

| Method | Ade Corpus V2 | Banking 77 | Neurips Impact Statement Risks | One Stop English | Overruling | Semiconductor Org Types | Systematic Review Inclusion | Tai Safety Research | Terms Of Service | Tweet Eval Hate | Twitter Complaints |
|---|---|---|---|---|---|---|---|---|---|---|---|
| T-Few | 80.4 | 69.5 | 83.3 | 67.6 | 95.0 | 91.5 | 50.8 | 73.6 | 75.0 | 58.6 | 87.9 |
| Human baseline [2] | 83.0 | 60.7 | 85.7 | 64.6 | 91.7 | 90.8 | 46.8 | 60.9 | 62.7 | 72.2 | 89.7 |
| PET [50] | 82.2 | 59.3 | 85.7 | 64.6 | 90.8 | 81.6 | 49.3 | 63.8 | 57.6 | 48.3 | 82.4 |
| SetFit [51] | 72.6 | 53.8 | 87.2 | 52.1 | 90.7 | 68.2 | 49.3 | 62.8 | 62.0 | 53.2 | 83.7 |
| GPT-3 [4] | 68.6 | 29.9 | 67.9 | 43.1 | 93.7 | 76.9 | 51.6 | 65.6 | 57.4 | 52.6 | 82.1 |

Table 11: Detailed per-dataset results for T-Few and the other top-5 methods on RAFT.