# OpenReview forum: "Few-Shot Parameter-Efficient Fine-Tuning is Better and Cheaper than In-Context Learning"
_NeurIPS.cc/2022/Conference — NeurIPS 2022 Accept_

### Official Review · Reviewer_EGYW · 2022-07-05

**Rating:** 5
**Confidence:** 3
**Soundness:** 3 good
**Presentation:** 3 good
**Contribution:** 2 fair

**Summary:**

This paper introduces T-Few, a parameter-efficient few-shot learning protocol that achieves STOA few-shot performance on many tasks, and also outperforms full-model finetuning and in-context learning with huge LMs. They rescale inner activations of LMs with learned vectors, such that only those vectors are updated during training. Their design is slightly different from existing parameter-efficient methods to enable mix-task batch training.
This paper demonstrates good engineering practice of parameter-efficient learning but there are flaws in the evaluations.

**Questions:**

.

**Ethics Review Area:**

["I don’t know"]

**Limitations:**

yes

**Strengths And Weaknesses:**

The proposed method demonstrates strong few-shot performance on popular benchmarks and challenging RAFT benchmark. Those engineering practices will be a good contribution to the community and serve as bases for feature research on parameter-efficient tuning.
The presentation is clear and easy to follow, and the motivation is well-justified and supported.

However, I have the following concern regarding evaluation.
Although the proposed method clearly outperforms baselines, I'm worried that the comparison is not fair. For instance, IA^3 is powered by two new training objectives and mix-task batch training. All these have been proved effective in improving few-shot performance. However, in the evaluation, those tricks are not applied on baselines (especially new losses), resulting in an unfair comparison.

There are also some mistakes in the description of the methodology, not sure if it's a typo or not. e.g., L212 we should minimize L_{LN} instead of maximizing it.

---

> ### Author Response · Authors · 2022-08-02
> **Answer to Reviewer EGYW**
>
> Thank you for your suggestions and questions. We have addressed each below and updated our submission accordingly.
>
> > However, I have the following concern regarding evaluation. Although the proposed method clearly outperforms baselines, I'm worried that the comparison is not fair. For instance, IA^3 is powered by two new training objectives and mix-task batch training. All these have been proved effective in improving few-shot performance. However, in the evaluation, those tricks are not applied on baselines (especially new losses), resulting in an unfair comparison.”
>
> Mixed-task batches is something that is possible with IA3 during during inference, but we don’t use mixed-task batches during training since the T-Few recipe has no shared trainable parameters between different downstream tasks and it would confer no practical benefits. Per your suggestion, we have run additional experiments of using the different training objectives across all PEFT methods and included them in our revision. Unlikelihood training and length normalization improve most of the methods, but IA3 is still the highest performing method. In addition, we note that all the results in Figure 2 have all the additional losses, so the comparison is fair.
>
> >There are also some mistakes in the description of the methodology, not sure if it's a typo or not. e.g., L212 we should minimize L_{LN} instead of maximizing it.
>
> Thanks for noting this. We have clarified in our revision that we maximize the length-normalized log probability by minimizing L_{LN}.

---

### Official Review · Reviewer_Fdzk · 2022-07-10

**Rating:** 7
**Confidence:** 4
**Soundness:** 3 good
**Presentation:** 3 good
**Contribution:** 3 good

**Summary:**

This paper presents a new parameter efficient fine-tuning (PEFT) method for pre-trained transformer models in NLP. The authors argue that, with a careful choice of parameters to be fine-tuned, this approach can be computationally much cheaper than in-context learning (ICL) and perform better than in-context learning approaches. The paper proposes a new method called IA3, where the trainable parameters are vectors that scale the activations in each layer. When applied to the T0 model, the authors show that the proposed fine-tuning method performs better than prior ICL and PEFT methods. In addition, a careful analysis is presented that shows the computation consumed by the proposed method and other approaches.

**Questions:**

* In Fig 2, how do you account for the difference in architecture, difference in pre-training data, etc.? Ideally, we would want to compare IA3 to other approaches for the same base model.

**Limitations:**

Seems adequate

**Strengths And Weaknesses:**

Strengths
* Clarity on experimental details, helps reproducibility.
* Extensive experiments and strong performance.
* Offers a practical recipe for few-shot learning.
* Clearly presented motivations.
* Computationally advantageous and better than in-context learning and other few-shot/parameter efficient learning methods.

Weaknesses
* The presentation can be reorganized so that the main contributions are highlighted better. For instance, IA3 could be introduced early on. The current presentation is dense and could use better structuring.
* Although fig. 2 presents an extensive comparison, it is unclear whether the authors accounted for the difference in model architectures, pre-training data, etc.

Overall this is an interesting paper with extensive evaluation of the proposed approach. I appreciate the careful analysis of computational requirements. The following issues have to be addressed by the authors.
* How were hyperparameters and other design choices made? Did you stick to the ‘true few-shot learning setting’, or were there any validation sets involved? I would recommend making this clear to make sure that comparisons with baselines are fair.
* Since the main method applies IA3 to T0, it is difficult to isolate the performance contribution of each. I would recommend presenting an analysis of IA3 applied to plain language models (no instruction tuning).
* Did you also try baseline methods (adapters, prefix tuning, etc.) with/without unlikelihood training, length normalization, etc.?

---

> ### Author Response · Authors · 2022-08-02
> **Answer to Reviewer Fdzk**
>
> Thanks for your thorough review and suggestions. We've updated our submission and responded below.
>
> > Although fig. 2 presents an extensive comparison, it is unclear whether the authors accounted for the difference in model architectures, pre-training data, etc.”
> >In Fig 2, how do you account for the difference in architecture, difference in pre-training data, etc.? Ideally, we would want to compare IA3 to other approaches for the same base model.
>
> All the results in fig. 2 correspond to different PEFT applied to the same backbone model with the same pre-training data, so there are no such differences to account for. We have updated our submission to make this more clear.
>
> > How were hyperparameters and other design choices made? Did you stick to the ‘true few-shot learning setting’, or were there any validation sets involved? I would recommend making this clear to make sure that comparisons with baselines are fair.”
>
> We use the validation set of the T0 held-out tasks to help design T-Few on T0-3B. However, we used the same set of hyperparameters across all tasks and did not extensively tune them. To test the true few shot ability of T-Few, we applied the T-Few recipe to T0-11B and RAFT (which has no validation set and is "true few-shot learning" by design) without any hyperparameter tuning. The fact that T-Few on T0-11B works so well on the T0 tasks and achieved SoTA and superhuman performance on RAFT suggests to us that our submission indeed satisfies the real-world requirements of few-shot learning.
>
> > Since the main method applies IA3 to T0, it is difficult to isolate the performance contribution of each. I would recommend presenting an analysis of IA3 applied to plain language models (no instruction tuning).
>
> Our primary goal in this paper was to present a single recipe (including a fixed model and set of hyperparameters) that worked well on any unseen few-shot NLP task. While we agree that it would be interesting to see how IA3 performs on other (plain) language models, we are interested in maintaining or focus on designing a specific recipe rather than developing a PEFT method alone. We would be interested in exploring applying IA3 to different models and developing improved PEFT methods in future work.
>
> > Did you also try baseline methods (adapters, prefix tuning, etc.) with/without unlikelihood training, length normalization, etc.?
>
> Thanks for suggesting this additional set of experiments. We have run these experiments and have added them to appendix D. SAID takes 3 days and so we weren't able to run all experiments during the weeklong rebuttal period, but they will be done in 2 days and can be included in future revisions.

---

### Official Review · Reviewer_yoAz · 2022-07-10

**Rating:** 6
**Confidence:** 4
**Soundness:** 3 good
**Presentation:** 4 excellent
**Contribution:** 3 good

**Summary:**

The paper focuses on few-shot learning in NLP and shows the limitation of few-shot in-context learning method (high computational cost and so on) and propose an adaptation strategy that adds light-weight learnable parameters (rescaling vectors) into a frozen pre-trained model and fine-tune only the learnable parameters on few labeled samples for few-shot learning. The experimental results shows the proposed strategy obtains much better performance than existing few-shot ICL methods and PEFT methods with lower inference computation cost while it requires largest training computation cost, disk cost and and higher memory cost during training T-Few.

**Questions:**

Some related literatures are missing [ref1,2,3].

[ref1] Requeima, James, et al. "Fast and flexible multi-task classification using conditional neural adaptive processes." Advances in Neural Information Processing Systems 32 (2019).

[ref2] Triantafillou, Eleni, et al. "Learning a universal template for few-shot dataset generalization." International Conference on Machine Learning. PMLR, 2021.

[ref3] Li, Wei-Hong, Xialei Liu, and Hakan Bilen. "Universal representation learning from multiple domains for few-shot classification." Proceedings of the IEEE/CVF International Conference on Computer Vision. 2021.

**Limitations:**

The limitations of the method are discussed.

**Strengths And Weaknesses:**

Strengths:

1. In overall, the paper is well-written.

2. The idea of reusing the pre-trained network and only learn additional light-weight parameters to adapt the pre-trained model for downstream tasks with few samples is interesting and shown to be obtain promising performance on downstream tasks.

3. The introduced rescaling vectors are light-weight.

Weaknesses:

1. One main weakness is the training cost of the T-Few. More specifically, when training the rescaling vectors on a new tasks, training T-Few will involves forward and backward passes through the whole model (though only rescaling vectors require gradients for updates) which will be computational costly for each tasks. Suppose there are a mass of tasks, the training cost is considerable.

2. Also, applying the proposed method to larger model can also increase the cost, e.g. applying to GPPT-3 with 175B parameters.

In overall, I think the idea is interesting and promising in the context of few-shot learning. However, I am not an expert on NLP field and I can not  assess the novelty of the paper in NLP.

---

> ### Author Response · Authors · 2022-08-02
> **Answer to Reviewer yoAz**
>
> Thank you for your detailed review. We have responded to your comments and questions below and have updated our draft accordingly.
>
> > One main weakness is the training cost of the T-Few. More specifically, when training the rescaling vectors on a new tasks, training T-Few will involves forward and backward passes through the whole model (though only rescaling vectors require gradients for updates) which will be computational costly for each tasks. Suppose there are a mass of tasks, the training cost is considerable.
>
> While the forward and backward cost could add up if there are many tasks to be fine-tuned on, the fine-tuning cost is only incurred once and will be amortized as the model is used more and more during inference, unlike ICL (Line 136). We found that the break-even point with GPT-3's computational costs occurs when GPT-3 has done inference on only about 20 examples. We think it's highly likely that most models will be used for inference on more than 20 examples. Table 1 and Section 4.2 on training costs both include additional information about the computational cost of computing gradients.
>
> > Also, applying the proposed method to larger model can also increase the cost, e.g. applying to GPT-3 with 175B parameters.”
>
> While the T-Few recipe was specifically designed to T0, we agree that it could be applied to larger models. We note that even with a larger model, using T-Few will be dramatically cheaper than using ICL since the inference cost will be k times smaller (where $k$ is the number of in-context examples). For example, if T-Few was applied to GPT-3, the inference cost would still be 32-times smaller for a few-shot dataset with 32 examples. We also would argue that applying T-Few to larger models may not be necessary given that already outperforms GPT-3 and while being able to be trained on a single GPU.
>
> > Some related literatures are missing [ref1,2,3].
>
> Thanks for pointing these out. We've added them to our paper.

---

### Official Review · Reviewer_jndu · 2022-07-12

**Rating:** 7
**Confidence:** 4
**Soundness:** 3 good
**Presentation:** 4 excellent
**Contribution:** 3 good

**Summary:**

The paper introduces a parameter efficient fine-tuning method that works for few-shot learning. The main contributions of the paper include: Extending T0 to work for few-shot learning. A single recipe for few-shot fine-tuning and a new parameter-efficient approach that outperforms prior work like prompt tuning and adapters.

**Questions:**

Please see above.

**Limitations:**

There is no discussion about limitations. For example, it is still an open question what makes AI3 to work better than methods like Adapters or LoRA.

**Strengths And Weaknesses:**

Strengths:

- The paper is extremely well written. It was a joy to read.
- The new parameter-efficient approach (IA3) achieves strong results compared with other methods.
- The paper achieves strong few-shot results with moderate language model sizes (up to 11B), outperforming more expensive models like GPT3.
- A single hyperparmeter and model configuration setting for fine-tuning on all tasks.

Weaknesses:

- Some of the ablations in the appendix do not show reliable improvements and standard deviation seems too high (assuming the subscript numbers in the the appendix table cells refer to stdev).
- Figure 2 shows the benefits of IA3 compared with prior work when fine-tuning T0-3B. But does the entire “T-few recipe” fail if another parameter efficient model (like adapters) is used instead of IA3? I.e., how important is AI3 within the proposed framework?
- I encourage the authors to move important ablations
 to the main text. If space is concern, related work could be shortened.
- The only ICL baseline considered is GPT-3. There have been several follow up works on improving ICL. More recent baselines like Chinchilla [Hoffmann et al., 2022] could have been also considered.

---

> ### Author Response · Authors · 2022-08-02
> **Answer to Reviewer jndu**
>
> Thank you for your review and suggestions. We have responded to your comments below and updated our submission accordingly.
>
> > Some of the ablations in the appendix do not show reliable improvements and standard deviation seems too high (assuming the subscript numbers in the the appendix table cells refer to stdev).
>
> Just for clarification, the subscript numbers refer to the interquartile range. We have made this more clear in the revision. While we do not always achieve a significant improvement on every dataset, we do consistently see a significant improvement when averaging across datasets. Removing pre-training decreases accuracy by 1.6%, removing unlikelihood training and length normalization decreases accuracy by 4.1%, and removing both pre-training and our additional loss terms reduces accuracy by 2.5%. Our goal in creating the T-Few was to make a general-purpose recipe that could be applied as-is to many datasets, and we believe the consistent average-case gains support this goal.
>
> > Figure 2 shows the benefits of IA3 compared with prior work when fine-tuning T0-3B. But does the entire “T-few recipe” fail if another parameter efficient model (like adapters) is used instead of IA3? I.e., how important is AI3 within the proposed framework?
>
> Thank you for making this interesting point. To clarify, fig 2 shows the performance for all parameter efficient methods. We have added an ablation of unlikelihood training and length normalization for all parameter efficient methods in appendix D. SAID takes 3 days and so we weren't able to run all experiments during the weeklong rebuttal period, but they will be done in 2 days and can be included in future revisions. However, swapping IA3 with other PEFT methods in the full T-Few recipe would involve pre-training each PEFT method. Since pre-training is computationally expensive beyond our means, we are unable to include this ablation.
>
> > I encourage the authors to move important ablations to the main text. If space is concern, related work could be shortened.
>
> Unfortunately, since the related work section is already in the appendix (Appendix B), we aren't able to include all ablation results tables in the main text due to space constraints. However, per your suggestion, we have included additional summary and emphasis of ablation results in the main text. While the gains are not always significant across datasets, we consistently do better on average across datasets.
>
> > The only ICL baseline considered is GPT-3. There have been several follow up works on improving ICL. More recent baselines like Chinchilla [Hoffmann et al., 2022] could have been also considered.
>
> In principle, we would love to directly compare with other few-shot ICL results from models like Chinchilla. Unfortunately, most large language models are not publicly available, and published results for non-public models (like Chinchilla) have only a small overlap with the held-out tasks considered by T0, T-Few, and GPT-3. Comparing on a small number of tasks would likely not be statistically meaningful. We hope that more publicly-available LLMs will be released so that more baseline methods can be compared to in future work.

---

### Author Response · Authors · 2022-08-02
**General Response**

Thanks very much to all of the reviewers for their constructive suggestions. We have responded to all reviewer comments and questions below and have updated our draft accordingly. Changes made include:

- Ran experiments on all the PEFT methods ablating the additional losses and added them to appendix D.  SAID takes 3 days and so we weren't able to run all experiments during the weeklong rebuttal period, but they will be done in 2 days and can be included in future revisions.
- Describe the ablation more in the main text section 4.4
- Clarified the introduction of the length-normalized loss
- Emphasized that we use the same recipe for all experiments after section 3
- Noted that the ingredients of our recipe don't always produce gains on every individual task but do consistently and significantly provide gains on average across tasks
- Added additional related work

If the reviewers have additional suggestions, we would be happy to incorporate them into an updated draft. Thanks again for your input.

---

### Meta-Review · Area_Chair_9qFK · 2022-08-26

**Recommendation:** Accept
**Confidence:** Certain

**Metareview:**

This paper demonstrates that Few-shot Parameter-efficient Fine-tuning (PEFT) is more accurate and dramatically less computationally expensive than in-context learning (ICL), and introduces a new PEFT method that varies the activity level depending on the learned vector and achieves high performance with only a few parameters. In addition, this paper proposes a simple way to apply it to the T0 model and shows that the proposed fine-tuning method performs better than the baselines. This paper is well written. The proposed method provides a simple and practical recipe for few-shot learning and shows strong performance on popular and challenging benchmarks. All three reviewers had similar positive comments on this paper. Thus the meta-reviewer recommends it for acceptance.

**Award:**

No

---

### Decision · Program_Chairs · 2022-09-14

Accept